# The Availability, Appropriateness, and Integration of Services to Promote Indigenous Australian Youth Wellbeing and Mental Health: Indigenous Youth and Service Provider Perspectives

**DOI:** 10.3390/ijerph20010375

**Published:** 2022-12-26

**Authors:** Janya Robyn McCalman, Ruth Fagan, Tina McDonald, Semara Jose, Paul Neal, Ilse Blignault, Deborah Askew, Yvonne Cadet-James

**Affiliations:** 1Jawun Research Centre, School of Health, Medical and Applied Science, Central Queensland University, Cairns, QLD 4870, Australia; 2Gurriny Yealamucka Health Service, Yarrabah, QLD 4871, Australia; 3Deadly Inspiring Youth Doing Good (DIYDG), Manoora, QLD 4870, Australia; 4Translational Health Research Institute, Western Sydney University, Penrith, NSW 2751, Australia; 5General Practice Clinical Unit, Faculty of Medicine, University of Queensland, Brisbane, QLD 4072, Australia; 6Apunipima Cape York Health Council, James Cook University, Smithfield, Cairns, QLD 4870, Australia

**Keywords:** First Nations, social and emotional wellbeing, co-design, service integration, adolescent

## Abstract

Concerns about the complexity, fragmentation and inefficiency of Australia’s current youth mental health service systems have led policy makers to seek improvements through a shift to community-based solutions. However, there is little evidence of how communities can make this shift. This paper examines the efforts of one Aboriginal and Torres Strait Islander (hereafter, respectfully, Indigenous) community—Yarrabah in north Queensland—to develop strategies for mental health and wellbeing service system improvements for school-aged youth (5–18 years). The research was co-designed with Yarrabah’s community-controlled health service and explores the perceptions of Yarrabah youth and service providers. Iterative grounded theory methods were used to collect and analyse data from 32 youth aged 11–24 years and 24 service providers. Youth were reluctant to seek help, and did so only if they felt a sense of safety, trust, relationality and consistency with providers. Young people’s four suggestions for improvement were access to (1) information and awareness about mental health; (2) youth facilities, spaces and activities; (3) safe and available points of contact; and (4) support for recovery from mental illness. Service providers highlighted an appetite for youth-guided community change and recommended five improvement strategies: (1) listening to youth, (2) linking with community members, (3) providing wellbeing promotion programs, (4) intervening early, and (5) advocating to address the determinants of youth mental health. Overall, both groups realised a disjunct between youth need and service provision, but a willingness to work together for systems change. This study demonstrates the importance of community-driven efforts that harness both youth and service providers’ perspectives, and suggests a need for ongoing dialogue as the basis for co-designing and implementing improvements to wellbeing supports and mental health services for Indigenous youth.

## 1. Introduction

The Australian government’s strategic framework for Aboriginal and Torres Strait Islander people’s mental health and social and emotional wellbeing (2017–2023) aspires to reach “the highest attainable standard of social and emotional wellbeing and mental health”. Achieving this requires mental health and related services to be “effective, high quality, clinically and culturally appropriate, and affordable” [1]. This framework is one of sixteen government policy documents about Aboriginal and Torres Strait Islander (hereafter respectfully termed Indigenous whilst acknowledging diversity) mental health, social and emotional wellbeing, and suicide prevention (2013 to 2018). Each policy cites an urgent need for new responses to address the increasingly high rates of Indigenous youth mental health issues [2]. They call for improvements to address the complexity, fragmentation and inefficiencies of mental health and wellbeing service systems, requesting systems that include the “best of both worlds” [3,4]. Systemic and transformative change is suggested through “a shift towards the community as the key place where mental health services and support are provided” [5]. 

To date, there is little evidence of community-based mental health and wellbeing services or how extant community-based systems and services could be improved to better meet the needs of Indigenous youth [6]. Australia’s mental health service system does not adequately address the complex socio-cultural factors that are faced by Indigenous youth in their attempts to access culturally safe treatment pathways [7,8,9]. This study takes a community-driven approach with an Aboriginal community-controlled health organisation (ACCHO) in Yarrabah, an Indigenous community in Far North Queensland. It engaged local Indigenous young people (aged 11–24 years) and service providers to explore their perceptions about the availability and appropriateness of existing community youth mental health and wellbeing supports and services; and to listen to their suggestions about how to improve the delivery and overall effectiveness of these services. 

There is an overrepresentation of Indigenous children and youth (0–17 years) who are hospitalised for mental disorders of 1.8 times the total Australian rate [10]. Mental disorders are defined as “a combination of abnormal thoughts, perceptions, emotions, behaviour and relationships with others” [11]. In Yarrabah, the ACCHO’s 2018 youth health check (ages of 13 and 25 years) identified 22 (18%) of the 122 youth participants (across sex and age groups) as having symptoms of depression ranging from moderate to severe [12]. However, in response to guidance from community partners, this study adopted a broad scope of focus from treatment services through to wellbeing and mental health promotion services and supports. Wellbeing is referred to as the holistic strengths-based concept of social and emotional wellbeing (SEWB). SEWB captures Indigenous people’s understandings and experiences of mental health, including the connection to body, mind and emotions, family and kinship, community, culture, country, spirituality and ancestry [13]. 

Indigenous youth are deeply concerned about mental health [13]. In a 2021 Australian national youth survey, Indigenous youth named their top three concerns as coping with stress (37%), mental health (34%) and body image (33%); with 44% of Indigenous youth reporting being stressed either all or most of the time [14]. Despite this, only 42% of youth reported that they would seek help for important personal issues from a GP or health professional [8]. Children under 15 years represent a third of the Australian Indigenous population, so improving health at this critical developmental period has the potential for long-lasting positive health and social impacts [15].

In redesigning youth health services, there is evidence of benefits from incorporating guidance from youth, based on their life experiences and knowledge of health services [16]. Youth-focussed research is a concept that has recently gained traction although is still not a mainstream research activity. Anderson-Butcher 2004 [17] proposed that there was a hesitation to trust and listen to young peoples’ ideas however, youth have the neurological and developmental capacity to engage [18]. Youth benefit from engagement with services through increased connectedness and social inclusion, contextualisation of experiences, and adherence to treatment [16]. Services also benefit through enhanced ability to connect and respond to youth needs, and increased program effectiveness and organisational credibility [16]. Furthermore, when service providers are involved in the redesign of services, organisational improvements are likely to meet with greater success [19,20]. Focussing on youth mental health, the 2013 Canadian Children, Youth and Communities Network report [21] proposed that youth mental health interventions were more effective when researchers, service providers and communities collaborated and shared their knowledge and resources. However, there is limited evidence of research or health programs that have incorporated Indigenous youth and/or service providers’ perspectives in the design or implementation of wellbeing and mental health interventions [6,22,23,24]. 

This aim of this paper is to explore how one discrete Indigenous community—Yarrabah—developed understanding to inform community-driven strategies to improve youth wellbeing and mental health supports and services. Yarrabah youth and staff of the primary healthcare service and other community based intersectoral organisations provided their perceptions about the availability, appropriateness and integration of wellbeing supports and mental health services for Yarrabah school-aged children and youth (5–18 years; hereafter termed school-aged youth); and, how these could be improved. The research questions were: How can wellbeing supports and mental health services for Yarrabah youth be improved? Sub-questions: (1) What contextual factors influenced the need for/provision of youth wellbeing supports and mental health services in Yarrabah; (2) what conditions were required to enable improvements in youth wellbeing supports and mental health services; and (3) the strategies suggested by Yarrabah youth and service providers to improve youth wellbeing supports and mental health services in Yarrabah? 

## 2. Materials and Methods 

The research was co-designed with the Yarrabah ACCHO, Gurriny Yealamucka Health Service (hereafter Gurriny) in response to their concerns about the level and appropriateness of mental health supports available to youth in Yarrabah. The research was established through a formal partnership agreement with Gurriny and conducted with the engagement, involvement, and leadership of Gurriny staff to ensure relevance, appropriateness, cultural safety, and usefulness. In Australian Indigenous communities, the ACCHOs such as Gurriny are the key health service providers. They provide comprehensive primary healthcare services including social and emotional wellbeing (SEWB) programs that incorporate community cultural resources and activities, Elders and families, and mental healthcare [25,26,27]. ACCHOs also coordinate with cross sector organisations to provide youth wellbeing support and mental health services. Services included in this study were primary and mental healthcare, education, youth programs and the State child protection and youth justice sectors. 

A modified community-based participatory research (CBPR) and social constructivist grounded theory approach were applied as the foundational research processes [28,29]. CBPR entails a cyclical process of exploration, knowledge construction and action. It applies the premise that the community is knowledgeable about their social realities and should be included in research activities—rather than simply being the subjects of research [28,30,31]. Constructivist grounded theory aims to construct theory from data, systematically obtained and analysed using comparative analysis from those who are “grounded” in a situation [32]. The research was governed by an Indigenous leadership governance group, four of whom are authors of this paper, including the senior author, who also leads the project governance group and two are from Yarrabah. In addition, research ethics were governed by the National Health and Medical Research Council (NHMRC) guidelines for the ethical conduct of research with Indigenous people and communities [33].

### 2.1. Study Setting

The Gunggandji and Yidinji peoples are the traditional custodians of Yarrabah, a community in North Queensland (for location, see Figure 1). Yarrabah is Australia’s largest discrete Indigenous community with 2505 residents at the 2021 Census, 98% of whom are Indigenous and with a median age of 25 years, compared to the median age of the overall Australian population of 38.4 [14]. The community, 50 km from Cairns, was established as an Anglican Mission in 1893, with Indigenous people relocated from other parts of Queensland [34,35]. 

In Yarrabah, Gurriny provides the generic primary healthcare service as well as a child and youth-focussed youth hub, and family wellbeing and family healing services. Queensland Health operate the Emergency Department and an outreach Child and Youth Mental Health Service. Other sector services available in Yarrabah are: youth bail support; an intensive family support program; individual child protection support; primary school based therapeutic counselling, primary and secondary (to year 10) school-based sports and aspirational sex-based programs and a Police Citizens Youth Club [36]. These programs are either funded and contracted by State government or operated by philanthropic organisations, often with little or no community consultation about need or implementation, and little integration between services [36]. Although these services provide much needed functions, there is significant demand for additional effective community-based services, particularly for those that encompass guidance from youth and community members.

### 2.2. Participants and Data Collection

This study applied CBPR methods through community yarning circles, Community Youth Advisory Group meetings and individual interviews. Yarning circles are often used to collect qualitative data in Indigenous health research settings as they are consistent with Indigenous approaches to consultation, ownership and ways of knowing. Yarning circles are conversational, non-hierarchical, and promote reciprocal sharing, active listening and learning [37]. Gurriny extended an invitation to community members and service providers, and youth to participate in two initial yarning circles in October 2019. 

The initial yarning circles with the youth (11–16 years), and with the service providers, were held at the Gurriny youth hub. They were facilitated by Gurriny Youth Wellbeing Officers and an experienced non-Indigenous researcher. Questions asked to both groups were: (1) how do you know if a Yarrabah school-aged youth is doing well, (2) what works in supporting youth wellbeing and mental health, (3) what is not working, and (4) what changes could be made? 

After the initial yarning circles, a Community Youth Advisory Group (CYAG) was established and older youth (16–24 years) were invited by the local Gurriny Youth Wellbeing Officers (based on their knowledge of the interests and capabilities of community youth) or self-nominated (through their engagement with Gurriny youth hub activities) to participate. The purpose of the CYAG was to seek guidance from older youth about their experiences of wellbeing supports and mental health services in Yarrabah that could inform improvements relevant to their younger school-aged peers (5–18 years). Establishing the CYAG addressed the concern that youth engagement is often ad hoc and included on a one-off basis [38]. 

Three CYAG meetings (see Table 1) were each attended by the Gurriny Youth Wellbeing Officer and facilitated by an experienced Indigenous youth leader affiliated from a youth empowerment organisation based in Cairns (Deadly Inspiring Youth Doing Good (DIYDG) https://diydg.org.au, accessed on 20 December 2022). Participants differed across groups. A researcher observed but did not participate in the meeting. The CYAG meetings lasted between two and two and half hours and CYAG members were given a AUD 30 gift voucher for their time and contribution. As advised by Blanchet-Cohen, McMillan [22], the CYAG meetings focussed on a small number of research questions. At the first two CYAG meeting, participants were asked (1) what youth wellbeing supports and mental health services exist in Yarrabah, (2) which supports and services are used by youth and are accessible, (3) do Yarrabah youth know about supports and services, (4) do you think that Yarrabah youth are respected by wellbeing supports and mental health services, (5) how can supports and services better support youth wellbeing, and (6) what resources are needed to make the supports and services in the community work better? 

Research findings summarised from the youth and service providers/community members to date were presented to the youth at the third CYAG meeting to obtain their feedback. Given the topic of discussion and the potential to trigger emotion, the Gurriny Youth Wellbeing Officer and yarning circle/CYAG facilitator provided participants with any follow-up support required. In total, 32 youth participated in the four youth consultations (Table 1).

The perspectives of service providers from the first yarning circle were augmented through individual follow up interviews (Table 2). These were facilitated by the Youth Wellbeing Officers, generally held in community although, during the 2020 COVID lockdowns, health restrictions were followed. Most of the participating service providers were Indigenous and were also Yarrabah community members. The youth and service provider research findings were presented at a second service provider yarning circle in June 2021. Gurriny staff and community service providers who participated in the initial yarning circle and/or interviews were invited to attend this yarning circle. The participants’ feedback was included in the final analysis. In total, 24 service providers participated in yarning circles and consultations (Table 2); 20 (83%) of these were also Yarrabah community members. Service providers worked for Gurriny (79%), the local school (12.5%), and other Yarrabah organisations (8.5%).

### 2.3. Data Analysis

Data from all yarning circles and CYAG meetings were audio-recorded and transcribed, and all personal identification was removed. Transcripts were imported into NVIVO version 12 software. Data gathered were analysed using grounded theory methods. Constructivist grounded theory methods followed an iterative process of data collection, and analysis using open, axial and selective coding to develop models of youths’ and service providers’ perceptions of mental health and wellbeing services and supports. 

Segment by segment open-coding to identify and name discrete concepts in the data commenced on receipt of the first yarning circle transcript [32]. The identified concepts shaped future data collection and analysis. In axial coding, new data from subsequent transcripts were compared to existing concepts for similarities and differences. Concepts with related meanings were grouped into categories [32]. The categories and the sub-categories were then integrated and refined through selective coding into two theoretical models that explained youth and service providers’: perceptions of the availability; capacity; appropriateness; and integration of services and supports. At feedback sessions, Yarrabah youth and service providers confirmed the resonance of the models.

### 2.4. Ethical Considerations

This research was designed as part of a five-year research project to inform the Yarrabah community-driven co-design of improved systems for promoting the mental health of Yarrabah school-aged youth [39]. Ethical approval was provided by Central Queensland University (0000021644), Queensland Education Research Inventory (550/27/2319), Queensland Health Cairns and Hinterland Hospital and Health Services (HREC/2021/QCH78083 (Oct ver 3)—1557), and Queensland Department of Children, Youth Justice and Multicultural Affairs—Child Safety (04458-2021) and Youth Justice (CYJMA 02361-2021).

After being informed about the purpose, aims and methods of the research project and prior to collection of any data, all participants provided written consent. For youth younger than 17 years, consent was also obtained from a parent or guardian. Youth participants were offered a prize for their attendance at yarning circles and a AUD 30 gift voucher for their participation at each CYAG.

## 3. Results 

Yarrabah youth and service providers had distinct perspectives of the core issue, contextual factors, and conditions needed to improve the availability, accessibility and appropriateness of youth mental health services and supports and suggested complementary improvement strategies (Table 3). However, at the interface between the perspectives of youth and service providers was a realisation of the disjunct between youth needs and the availability of appropriate supports and services, and an expressed desire to work together collaboratively for systems change. 

### 3.1. Core Issue: Realising a Disjunct and Desiring to Work Together for Systems Change

As depicted in Table 3, Yarrabah youth and service providers both acknowledged that the current Yarrabah service system was not working well to support youth wellbeing. Service providers identified a need for youth guidance in planning and implementing improvements to the service system, and youth were willing to provide that guidance. Lacking in the current Yarrabah service system, however, were clear mechanisms for youth leadership capacity development or youth advisory structures for improving service systems. To inform the improvement of youth wellbeing systems going forward, Yarrabah youth and service providers identified contextual issues, conditions and strategies that pertained to different levels of focus within the community environment: individual youth, families, services, and service systems.

#### 3.1.1. Youth: Reluctance to Seek Help

At the core of the youth narratives was a feeling of reluctance to seek help. They spoke of a community expectation of youth resilience and strength as a double-edged sword that had an affirmative aspect but also led to reluctance to acknowledge wellbeing concerns. One youth explained: “*There’s a lot of terms that go around in Yarrabah, like ‘oh you gotta be strong’, ‘be strong for your family’, ‘be strong for this person, that person’ but there’s nothing that actually… like addresses it*.” All youth participants reportedly had friends who had struggled with mental health issues, but they were hesitant to talk openly about it. One said: *“it’s not an open conversation sort of thing… It is a concern I think…”* Another youth suggested: “*I think mental health should be talked about because people need to know what’s going on*.” Youth were consequently reluctant to access services unless they felt comfortable with the provider. One participant commented: “*you’re going to talk to a stranger… you don’t know the person… you’ve never met them and then you’re just going to talk about big problems and it’s just a stranger*.” Youth suggestions for improving the service system in Yarrabah included increasing information and awareness about mental health; providing safe and available points of contact; access to youth facilities, spaces and activities; and supporting recovery of youth with mental illness. Youth were willing to provide guidance to service providers in ongoing systems improvement efforts.

#### 3.1.2. Service Provider: Appetite for Youth-Guided Community-Driven Change

At the core of the service providers’ narratives was an appetite for youth-guided community-driven change. They acknowledged that current service delivery was not working well, commenting: *“Yarrabah population is really young and it just… sort of baffles me that we… have not been able to get it right in this community—the actual model for our youth.”*


Service providers recognised the value of youth guidance in designing youth services. One suggested: “it takes a community to raise a child so that’s what we come together and start doing. But like you said do it their way, their voice, not what we think it is—right or wrong. All we just there for is our guidance”. Another commented: “If we don’t… allow them to advocate for their rights, we’re doing it for them, and that’s what government has done for us for many, many, many years. So let’s flip it on its head and make it that it’s our young people talking… We’re empowering them to have that voice and really, truly acknowledge the social justice principles of equality, access, and their right to participate”. Service providers suggested improvement strategies as listening to youth, linking with community members, providing wellbeing promotion programs, intervening early, and advocating to address the determinants of youth mental health.

### 3.2. Individual Youth Level of Influence

At the level of influence of individual youth concerns, youth perceived the contextual factors that affected their wellbeing and use of supports and services as being related to their hiding mental health concerns. The contextual factors identified by service providers focussed on intergenerational and personal trauma. Enabling conditions for improvement were perceived by youth as safety, trust, relationality and consistency. Service providers similarly considered the enabling conditions as workforce cultural safety. The improvement strategies identified by youth were having information and awareness about mental health. Service providers suggested listening to youth as an improvement strategy. 

#### 3.2.1. Context

##### Youth: Hiding Mental Health Concerns

Yarrabah youth perceived that community members had an expectation of youth as resilient and strong. One youth stated: “Like your family’s willing to help you—it’s just that people think they’re so independent and can do it on their own. They’re ashamed to open up”. Youth recognised that the silence around wellbeing and mental health started at an early age and discouraged youth to seek help: “At a very young age we don’t really take care of our mental health, it’s not something that we [are] a hundred percent on board with…” Another added: “there’s just no conversation around it when you’re young. You just don’t know about it ‘til you’re older.”

When asked why young people were reluctant to seek help, CYAG participants responded in unison, “shame”. One youth elaborated: “It’s that vulnerability thing. You’re laying yourself open to judgement. And you really shouldn’t be but that’s what happens. Coz people don’t understand it, so they don’t like it if they don’t understand it. They feel like that stigmatised view… Most people see depressed people wanting to not do anything and so they don’t understand it’s more complex than that.” One youth recalled his friend’s attempt to get support: “one of the fellas was going through something and like he has a lot of friends, a lot of family networks and all this and that but… they won’t all come together and they won’t address how they’re feeling.” 

In consequence, mental health was not openly spoken about in Yarrabah, with one youth stating: *“Yeah it [mental health]’s a concern but no-one recognises it as a concern. It’s just something that gets swept under the rug”*. Mental health concerns were kept hidden and early intervention support or services were rarely accessed. One youth reflected: “*People aren’t really out in the open when it comes to this sort of stuff like mental health, like nobody wouldn’t want to talk about, ‘I’ve got anxiety’ or ‘I’ve got depression,’ like out in the open*.” 

In the absence of support, for some, the effect of unresolved mental health issues was often emotional dysregulation and/or self-medicating with alcohol or drugs. A youth observed: *“People do like to express feelings… like when it gets too much and they have no one to talk to, they just run to social media and just explode.”* Another youth observed: “*They think it’s alright just to keep it to themselves. But it actually is a bad thing until they all bungle up and then end up going mad when they’re drunk or whatever*”. Another youth said: “*Follow the leader. Like alcohol and drugs… People think that it will help but it doesn’t in the long run. It destroys it quick fast*”. Another added: “*That’s the hard truth. Most youths here will turn to alcohol before they deal with the situation in front of them*.”

##### Service Providers: Seeing Intergenerational and Personal Trauma

Service providers corroborated youth perceptions that community adults expected youth to be strong. They acknowledged youths’ strengths such as vision and purpose, a sense of spirituality and belonging, willingness to act as role models for others, connectedness with friends, and engagement in education or employment. One said: *“you hear a lot of bad stuff that happened with the young people, but I see a lot of good things too—young boys looking out for their mates who might not have the same things*”. Another observed: *“You see them sitting down in their little circles having yarns and that laughter. I think humour for our people plays a major role in that stress release.”*

Unlike youth, service providers spoke of the effects on wellbeing of historical and contemporary processes of colonisation that have disrupted traditional family and kinship structures, child rearing, and cultural knowledge and practices in Yarrabah. A service provider and community member said: *“It’s not just what’s going on now, that’s trauma from our great, great grandparents. That’s still being passed on. The lack of cultural connection, the lack of identity, even it’s not knowing our language… that was stolen from us…”* She continued: *“we really only have six-seven generations here in Yarrabah, and over those generations there’s been a lot of trauma transferred… that loss of culture, loss of direction, that trans-generational trauma… has played a major role in where our kids are now, today”.* Another service provider explained the need to “*acknowledge the past to begin the healing and begin the process of moving forward. Without that knowledge in the past, we’re lost. We don’t know where we’re going because we don’t know where we’ve come from*”.

Service providers also identified the risks to wellbeing of contemporary personal trauma, including cumulative loss and grief, child abuse, bullying and use of alcohol and drugs. A Yarrabah service provider spoke of the cumulative impact of grief in response to the many deaths experienced in the community. She recalled: *“over a couple of months we’ve had seven people pass away in this community alone… from a young person right through to elders. What we fail to remember often… is check in on our young people. ‘How does that affect you as a young person?’, how do these young people grieve? We see that through the school gates, the behaviour is escalating constantly.”* The service provider elaborated: *“We’re expecting them to keep on, keeping on and participating in that which triggers the mental health issues. They don’t know where to go with it sometimes. Or there’s too much of it and how do they manage that? How do they internalise that?”*

Bullying was another contributor to personal trauma identified by a service provider: “Bullying from social media. There’s a lot of that going on… That’s a real impact on mental health in young people”.

Service providers considered that alcohol and drugs were a consequence of diminished wellbeing and another important risk factor for mental illness. A service provider observed: *“Over the last couple of months, that’s what we’ve seen, that behavioural acting out of, ‘I don’t know what to do.’ ‘I’m tired.’ ‘I’m angry.’ And then they gravitate to the alcohol. They gravitate to the drugs*.” 

#### 3.2.2. Enabling Conditions

##### Youth: Safety, Trust, Relationality and Consistency

Feeling trust and a sense of safety was a pre-requisite to help seeking youth. CYAG participants identified the qualities of interactions that they valued from services, including: *“safe and connecting, trustworthy; and caring; respect; yarning up with friends and family; safety; love, truth, laughter, company connection.”* Such descriptors support a clear need for workers to have strong listening skills, empathy, and communication skills. 

Youth were reluctant to use face to face services unless they felt comfortable with the provider. One participant recalled: “I used to go to headspace [youth mental health service] before but… I don’t find it comfortable because I don’t really know them. That’s why I prefer coming over here.” Youth also expressed a deep mistrust about the safety of some services. One said: “A lot of people still look at the police as they look at doctors you know? Like going to hospital because you’re going to die… That’s the same as a lot of people don’t call police because of the brutality stuff and that so.” Another said: “Not everybody feels safe going to a doctor because that’s a white man.“

Youth participants considered that it was important to have a relationship with the person they were trying to talk to. One youth related that his cousin would engage with the local men’s health or youth health worker, but the initial disclosure of his concern would be tentative and indirect: *“They won’t bring it straight out. A person like [men’s health worker or youth health worker] will have to pull it out of them. And like it won’t be in a group… it’ll have to be one-on-one with the man. They won’t bring it out in front of their friends or family or whatnot. It’d have to be someone like [worker’s names] and then go for a drive or go down to the Men’s Space.”* Another disclosed her well-considered plan: *“If I went through- you know like anxiety… I’d just search stuff up on the internet… and then probably talk on the phone with them. I’d like to get two people opinion. Like I talk to one person that I know because they know what I go through and all this and that and my background, and then talk to a person that I don’t know, like who doesn’t know me in my community.”*

Another youth agreed that the anonymity of a professional contact could be protective. Referring to his reluctance to access services, he said: *“Everybody knows everybody– and sometimes I just don’t want to say my name.”* Another explained: *“I would rather talk to my family and friends but at the same time, I don’t want to put that burden on them or I don’t want them to look at me like that—like you know? Business hours, I’d probably just go and talk to a doctor that understands—and they won’t look at you like—they’ll speak to you the same way”*. However, they considered that workers needed to have experience, good listening skills and longevity of positions and therefore consistency in relationships. One reflected: *“Some people just give that hand for probably a day and that person gone again. And then that person who going through something, they by themself again so… so nobody deals with it properly.”*

##### Service Providers: Workforce Cultural Safety

Service providers also identified a need for cultural safety through increased understanding by health professionals of the historical and cultural influences on Indigenous children’s and youth mental health, and improved practice using trauma-informed approaches. They noted a need for improved attitudes towards and relationships with Yarrabah youth. One said: “*One thing I think has stood out for me in Yarrabah is the disconnect with the Police. There needs to be such a relationship built up with the Police and young people, because all they see is Bully Man. And what I’ve found and what I’ve heard so there needs to be such a drive by Police to build that relationship up that we’re not just the tough people that criminalise you*”. 

#### 3.2.3. Improvement Strategies

##### Youth: Having Information and Awareness about Mental Health

CYAG participants suggested an increase in information about youth wellbeing and mental health in Yarrabah. A youth said: “*I think there’s not a lot of education about it… It’s not addressed*.” Young people suggested three methods of raising awareness of mental health concerns and promoting wellbeing and mental health: through face-to-face education by role models, social media, and tailored youth specific information. 

Youth participants suggested that community events and/or on-country workshops could be used to promote mental health awareness training and education. One suggestion was to have: *“mental health week in the community where we can run games and promoters to come in—them big ones that talk at the big youth events… it will open a lot of our young men minds up too.”* Another suggested that community radio could be used *“like* a *news program where you have guests on that can talk.”*

Youth also suggested social media for youth wellbeing and mental health concerns and support. A youth suggested: “the Youth Hub or Youth Council can run the Facebook page and target the audiences. Not just a Facebook page. But Snapchat, Instagram.” Another suggested “putting good news stories of actual youth in our community and we can also tap into the Yarrabah Newsletter…The youths in the community, they’d feel good about theirself—then they’d see things up on social media.” Youth also suggested the development of a Yarrabah youth mental health app: “Even interactive stuff… Say a question could be, ‘if your friend comes up to you and tells you this, what would you do?’ And you click, and it takes you down the pathway—Yeah. Even if you do it wrong or you don’t like that answer, you can go back and choose a different path. Instead of actually doing it in real life if it happens.” Another concurred” If they’re feeling down, they can just go to the app and start talking to them people and whoever’s behind the app can talk to them there too. Gotta be anonymous.”

Finally, youth considered that there was a role for youth-specific information, for example pamphlets promoting youth mental health awareness. Referring to information about depression and anxiety, a Yarrabah participant suggested: “*Having something like a map or a pamphlet or something with frequently asked questions where they could just read through… But it’s gonna be a short and sharp stuff. No big paragraphs and in our slang, like in the way we talk. For us in Yarrabah, we’re short, quick*.”

##### Service Providers: Listening to Youth 

Service providers recognised the importance of listening to youth for guidance about improvement in youth wellbeing services. One recognised: *“As an older person in the community, I’m really out of date and out of touch with what will work for Yarrabah youth.”* Another said: *“From a service delivery point of view for us is to really try and listen to what is coming out of the youth voice—we’ll really do an injustice if we don’t take that on board and expect it could potentially be really completely different to what we have in our mind and may require some significant shifts in the way we deliver services.”* Another local service provider advocated: “*I’m a grandmother. I have three children—grand-children, whom I love to death… Give me a headache [laughs] but they’re my future… If we can’t get it right for them, then I hate to think where we’re gonna be. Because we’ll die, and the government of the past would’ve succeeded in killing our people off*.”

### 3.3. Family Level of Influence

At the level of influence of the (extended) family, youth identified the key contextual issue as informal support from families but did not identify enabling conditions to facilitate improvement. Service providers identified the key contextual factors as families as role models and connectors to culture. Enabling factors identified by service providers were lining with community members. Youth identified the key strategies as having safe and available points of contact, whilst service providers identified building community capacity.

#### 3.3.1. Contextual Factors

##### Youth: Informal Support from Families (or Not)

CYAG participants acknowledged that in many families, members provided important and positive informal support for youth wellbeing. One youth said: *“family’s a big thing in Yarrabah. It’s pretty much everything. It’s your support, guidance. It’s where you find your cultural identity and all that kind of stuff.”* Extended family and peer supports were commonly mentioned as their first option for talking about emotional concerns. A youth commented: *“Like for me personally I would be, ‘go see this person’ type thing. Or someone that’s older, like an older adult. Someone I know I can trust”*. However, youth also recognised that not all families were as supportive. One youth stated that *“Most young kids don’t have relationship with their parents. And they don’t have someone to tell it to, too. Like a safe place to talk about it.”* Young people were concerned that some family members might judge young people for having mental health concerns and respond inappropriately: *“that judgement you know? You see that in your family unit so you don’t want to talk about it. Like a catch twenty-two kind of… yeah it’ll come out. So use it against you.”* When asked where young people would go if they needed help, a Yarrabah participant said: *“Their friends. Yeah, they wouldn’t talk to their parents.”*

##### Service Providers: Family as Role Models and Connectors to Culture

Service providers reflected that community members contributed as role models to support youth wellbeing. One service provider considered that “*it’s all up to the adults to set a good example for the young generation. That’s the reason why I’m trying to set a good example for our generation. I was a rugby league player and through that, I tried to set the example… by not drinking*.” Service providers also identified that young people were connected to their cultural roots through families, and that this was an important protective factor.

#### 3.3.2. Enabling Conditions

##### Service Providers: Linking with Community Members

Service providers (many of whom were also community members) also noted the value in closer links between services and community members in relation to better supporting child and youth wellbeing.

Service providers suggested a need to link with community members to strengthen community and cultural cohesion. A Yarrabah service provider suggested: *There shouldn’t be just a big blow-up for a NAIDOC Day. That should be every day. It should be settled in your culture every single day, not just one day a year.”* Yarrabah service providers grappled with how to link culture with youth wellbeing: *“How do we work with them to be able to maintain some of that cultural value and the principles, but still live in this modern world? Because technically even now, our kids still have one foot in the traditional camp—Aboriginal cultural camp, and the other one in the white cultural camp.”* Another service provider said: *“Building the resilience… bringing them together to actually look at how we can actually work together and become a community. Coz a community’s what keeps us strong.”*

#### 3.3.3. Improvement Strategies

##### Youth: Safe and Available Points of Contact

Yarrabah youth suggested identification of and training for volunteer community gatekeepers to extend their current frontline connections with adults to providing safe connections for youth. Participants suggested that having younger and gender-matched Indigenous workers would make a difference to the initial relational interchange between support workers and Indigenous youth. A Yarrabah participant suggested: *“Even if it’s a step-by-step thing, so you might have the younger person there to connect with you… and then the older one come in with more experience and different advice… like if I had girl problems, I’m not going to talk to him*… *even having someone like L-G-B-T-Q-I in it as well.”* Youth suggested that these trusted contacts could be identified trained in mental health first aid. One youth suggested: *“Only people who’s hearts are in the right spot and want to help out… so I feel like there should be some people with mental health first aid, suicide prevention training—some sort of idea.”* Another added: *“something like a logo… an icon, or a symbol, posted up everywhere, on shirts… And sort of like identifying someone without having to ask… then they can be the ones to say ‘are you alright?’*”

##### Service Providers: Building Community Capacity

A service provider described the potential of extending community gatekeeper training to also focus on youth: “We try and upskill a lot of our community members in mental health and first aid so when they become first responders, we do have mental health supports, but they’re not based with youths. They send someone over from Cairns every month or so for the youth side of it.”

### 3.4. Service Level of Influence

At the service level of influence, youth identified the key contextual factors as connecting with support and services (or not). Service providers perceived the key contextual factor as the capacity of services and supports. The key enabling conditions were identified by youth as the adequacy and appropriateness of services, and by service providers as workforce capacity and sustained resourcing. Strategies identified by youth were having access to youth facilities, spaces and activities, and supporting the recovery of youth with mental illness. Strategies identified by service providers were providing wellbeing promotion programs and intervening early.

#### 3.4.1. Context

##### Youth: Connecting with Supports and Services (or Not)

Youth participants had some experience of using local face-to-face wellbeing promotion programs through schools and community youth events. When asked what facilities or activities they had used for mental health support or well-being, a young male participant said: “*For my mental health? What happens to me? I go to gym training. PCYC… If I’m stressed out, that’s where I go*.” Another offered: *“I go Men’s Space when I feel down*.” However, they commented that there were few services available, and those that existed were limited in terms of their capacity to meet the needs of Yarrabah youth: *“They should put disco back on… There’s no movies here. No movie night! They don’t do events on Christmas*.” Another said: *“Police Citizen Youth Club…They try to get kids off the streets at night and they do something with them for I think two hours then send them back home… they should do that for girls!“*

Young people had also used mental health services through the headspace service and hospitals but reported feeling that the support was inadequate for their needs. A Yarrabah participant disclosed: “*I had like a sorry business. I had like a little episode? And when I went into Cairns Base [hospital]… it just frustrated me that I went and talked to this person and then another nurse came in and then I had to repeat my story… if I were to have another episode like I did, I probably wouldn’t want to go back. Having to tell the same story within a half an hour to three different people, that was just… exhausting*.”

##### Service Providers: Capacity of Services and Supports

Service providers concurred with youth that services to support youth mental health were largely unavailable. A service provider noted: “*With Gurriny you’ve only got… two [peer] youth workers… and probably annually they have that Youth Forum but other than that, nothing with youth mental health*.” Another said: *“mental health… in Yarrabah as well, for young men, it’s something that’s swept under the carpet. You don’t really address it.”*

Service providers identified the need for advocacy to improve the social and economic determinants of Yarrabah youth wellbeing. They suggested: “*focusing in on social determinants. Like their employment and housing… They all kinda lead to one another so it’s kinda like a circle*.” Another suggested that community organisations could consider the housing needs of youth in more flexible ways: “*Ideas around single accommodation… Units for small families or- inter-generational houses that accommodate a whole range of ages or families who want to live together*.” One service provider noted the benefits of employing youth: “*We paid five workers… and it’s the best thing coz we get to mentor them and start them up in the job, but they also get to feel like what it’s like to be employed*.” Another considered that encouraging youth to leave the community for employment was beneficial: “*I think it’s around changing the mindsets of our young people that you can go… and then come back. Look at this as a place of retreat or of coming back to ground yourself culturally*.”

Service providers suggested that such broad strategies require intersectoral coordination. One noted: “I’m a strong believer in utilising the resources that we have available to us… and I think it comes back to that coordination… it’s about building that cultural realm around who we are and where we come from, and the strength that comes with that. Which is really important, but how do we start that conversation? How do we start to actually get that happening in our community?”

## 4. Discussion

Indigenous communities and leaders have called for local leadership of responses to youth mental health problems to ensure that they address both cultural elements, lived experience, and respect for Indigenous people’s right to self-determination and governance in matters that impact upon them [4]. This study has attempted to respond to this call by engaging Yarrabah youth and service providers to develop place-based responses to inform youth mental healthcare improvements as the first stage of youth-guided and community-driven efforts to advance wellbeing supports and mental health services for Yarrabah youth.

At the core of youth and service providers’ perspectives about how services and supports for youth wellbeing could be improved, there was realisation of a disjunct between need and supports/services, but a desire to work together for systems change. There were clear synergies between the youth and service provider perspectives, but also important differences (Figure 2). The contexts identified by youth included hiding mental health concerns, informal supports from families (or not), connecting with services (or not), and dealing with family, home and community environments. Service providers identified that demand for wellbeing supports and mental health services was driven by seeing intergenerational and personal trauma and family socio-economic stressors, and that they were concerned about the availability and capacity of services to respond to youth needs and service fragmentation. Service providers acknowledged that that some families provided informal support as role models and connectors to culture.

As has been found elsewhere, Yarrabah youth were reluctant to seek help [26,40]. Instead, they hid their mental health concerns. Settipani, Hawke [40] found that non-Indigenous youth also experience mistrust with mental health programs, who deemed some as being detrimental to their health needs. However, the Yarrabah youth participants in this study expressed an even greater need for trust building, relationality and consistency of services and supports because of the complexity of historical and contemporary trauma and socioeconomic contextual factors they experienced [26]. Yarrabah service providers acknowledged that they needed to hear youth perspectives to inform youth-guided community-driven change.

From the perspective of the youth, the presence or absence in services of cultural safety was considered to entail trusting and respectful connections with workers with experience and good listening skills, where they felt cared for, and where laughter was shared. This finding was also noted in a study from Mt Isa, where trusting interpersonal relationships between Indigenous youth with health practitioners were found to underpin the value of all improvement strategies [41]. Literature reviews by Walker and Reibel [42] and Settipani, Hawke [40] also found that Indigenous youth preferred youth-friendly services that were known within their social circles or referred by their peers. The things youth valued in a service were friendship and a safe place to talk with trusted youth workers about their health issues [7], whereas negative attitudes and stigma posed a barrier to the detection and treatment of depression [27]. Yarrabah service providers identified similar conditions of workforce cultural safety [43,44]. Both youth and service providers also acknowledged the importance of the relational nature of service provision—although youth identified the importance of relationality from service provider to client whereas service providers focussed on the relationality of providers across services. The key strategy suggested by youth for supporting greater openness about mental health issues was the promotion of information and awareness about mental health. A lack of information and awareness about mental health has been reported elsewhere as a barrier to youth mental health service use. For example, a systematic literature review by Radez, Reardon [45] found that 96% of studies reported a limited knowledge of mental health and broader perceptions of help-seeking by youth and 92% studies reported perceived social stigma and shame. Yarrabah youth were unaware of the generic Indigenous youth mental health apps that already exist [46,47] and considered that the unique contextual factors found in Yarrabah required a tailored approach to social messaging [48].

Youth and service providers considered that families played an important role in providing informal support, role modelling and connection to culture. Service providers considered that they could link more closely with community members, and Yarrabah youth suggested the implementation of a system of safe and available points of contact/linking with community members. Such gatekeeper systems have been provided in other Indigenous communities, with training provided to community contact people, and has resulted in increased knowledge about suicide, confidence in identifying people who were suicidal, and intentions to provide help [49]. 

Both youth and service providers were concerned about the capacity of services to provide for the needs of Yarrabah youth adequately and appropriately. Service providers considered that workforce training and sustained resourcing were the key enablers to improvement. Suggested strategies were access to youth facilities, spaces and activities, wellbeing promotion programs, and supporting the recovery of youth with mental illness. Youth suggested online booking and/or service provision as options for enhancing confidentiality and mitigating any shame associated with a mental health consultation [50]. Youth participants spoke of the need for youth activities and for youth-specific spaces and facilities—there is evidence that community-based, integrated youth service hubs can create a single point of entry to comprehensive services and have the potential to make wellbeing supports such as counselling more accessible and to coordinate intersectoral approaches [40,51,52]. Similarly, youth prioritised the strategy of supporting recovery of (other) youth with mental illness. This demonstrates the concern and commitment of Yarrabah youth to support the wellbeing of other youth, and a sense that they needed to take responsibility for providing daily support. Whilst admirable, such support requires the backing of professional health service providers. The strategies suggested by service providers but not youth were intervening early and advocating to address the determinants.

Finally, youth and service providers recognised the impacts of the broader determinants such as housing issues, economic prospects, and family income on youth mental health. For youth, these affected their participation in services and activities, for example the cost of sports, arts and other youth activities was a barrier to participation. Service providers were clear that no single service had the capacity to address these determinants, and that a stronger community-based collective approach was needed.

The pragmatic and often low-cost improvements suggested by youth were fed back to service providers to inform the co-design of novel youth-guided and community-driven ways to support mental health services for Yarrabah youth wellbeing. Importantly, Gurriny was able to consider implementation of the suggested strategies as the research progressed. For example, Gurriny youth wellbeing staff attended a meeting of the Yarrabah Leaders’ Forum, a collective of leaders from community-based services in Yarrabah, and were invited to create a youth-run Youth Council, and social media messaging was developed by members of the Yarrabah CYAG. Gurriny has committed to implementing an ongoing youth advisory group to work with service providers and to contribute to the ongoing research process. Further research is needed to explore how well Yarrabah youth and service providers work together to inform the co-design and implement improvements to place-based youth wellbeing supports and mental health services, and to evaluate the outcomes. Priorities include Yarrabah youth leadership and empowerment, workforce training in trauma-informed care, and healthcare systems approaches to enhance the availability and capacity of preventive wellbeing programs, early intervention, and recovery from mental health disorders.

### Limitations

Whilst a range of youth and service providers participated in the study, COVID-19 lockdowns and travel restrictions limited the extent to which face-to-face data collection methods were feasible. This paper is therefore based on yarning circles, CYAG meetings, interviews with service providers, and the feedback from ACCHO Youth Wellbeing Officers present at youth yarning circle and CYAG meetings that could be arranged as was feasible. The findings are specific to the situations in Yarrabah and would need to be adapted for other situations and settings.

## 5. Conclusions

This research responds to policy calls for Indigenous community-driven initiatives to improve extant mental healthcare for Indigenous youth. Reluctance to seek help was identified as the core issue for youth, while an appetite for youth-guided, community-driven change was the core issue for service providers, who were acutely aware of large areas of unmet need and failings in the way mental healthcare services are provided. Bridging the perspectives of youth and service providers was the realisation of a disjunct between youth needs for support and the availability and capacity of services, but a willingness to work together for systems change. Improvements recommended by both youth and service providers were: (1) promoting awareness, (2) listening to youth, (3) providing points of contact in the community, (4) building community capacity, (5) providing youth-specific spaces and activities, (5) providing wellbeing promotion programs, (6) early intervention, (7) supporting recovery, and (8) advocacy for improvement in the broader determinants. The study demonstrates the importance of harnessing both youth and service providers’ perspectives, and ongoing dialogue as the basis for co-designing improvements to wellbeing supports and mental health services for Indigenous youth.

## Figures and Tables

**Figure 1 ijerph-20-00375-f001:**
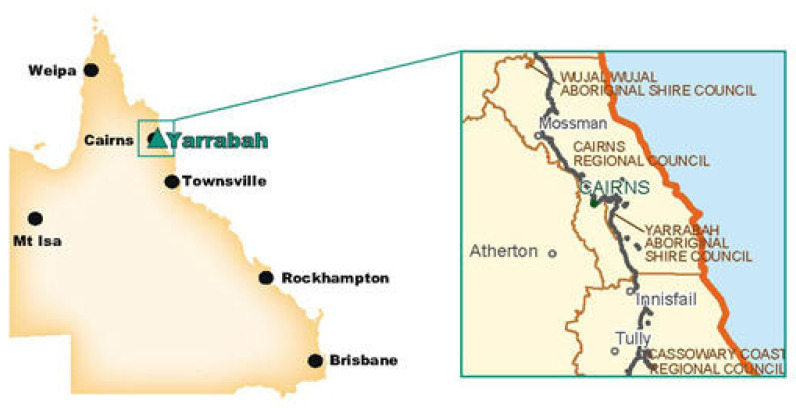
Map of North-East Australia, highlighting the location of Yarrabah.

**Figure 2 ijerph-20-00375-f002:**
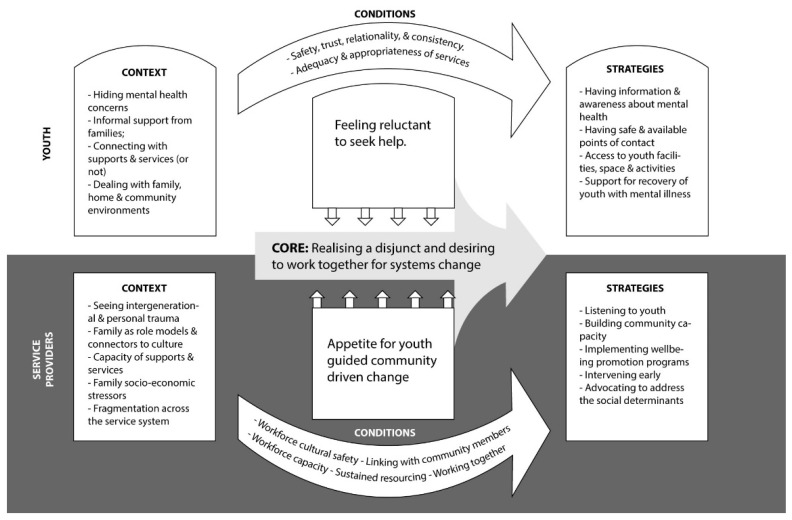
Youth and service providers’ perspectives of Yarrabah youth wellbeing support and services.

**Table 1 ijerph-20-00375-t001:** Youth participants.

Yarrabah Youth Meetings	Number of Participants	Age Range (Years)	Sex	
			Female	Male
Youth yarning circle, October 2019	12	11–16	11	1
CYAG 1, October 2020	6	18–24	2	4
CYAG 2, January 2021	9	16–24	5	4
CYAG 3—Research feedback May 2021	5	16–24	3	2
TOTAL	32	11–24	21	11

CYAG = Community Youth Advisory Group.

**Table 2 ijerph-20-00375-t002:** Service provider participants.

Yarrabah Service Provider Participants	Number of Service Providers	Indigenous	Non-Indigenous	Health Sector	Other Sectors
Yarning circle 1, October 2019	10	8	2	8	2
Community Service provider interviews February 2020–March 2021	9	8	1	7	2
Yarning circle 2—Research feedback June 2021	5	4	1	4	1
Totals	24	20	4	19	5

**Table 3 ijerph-20-00375-t003:** Socio-ecological framework for improving Yarrabah youth wellbeing supports and services: Comparison of service provider and youth perspectives.

Level of Influence	Perspective	Context	Enabling Conditions	Improvement Strategies
CORE	Realising a disjunct and desiring to work together for systems change Youth core: Reluctance to seek help Service provider core: Appetite for youth-guided community-driven change			
Individual youth	Youth	Hiding mental health concerns	Safety, trust, relationality and consistency	Having information and awareness about mental health
	Service provider	Seeing intergenerational and personal trauma	Workforce cultural safety	Listening to youth
Families	Youth	Informal support from families (or not)	None reported	Having safe and available points of contact
	Service provider	Family as role models and connectors to culture	Linking with community members	Building community capacity
Services	Youth	Connecting with supports and services (or not)	Adequacy and appropriateness of services	Access to youth facilities, space and activities; support for recovery of youth with mental illness
	Service provider	Capacity of supports and services	Workforce capacity; sustained resources	Implementing wellbeing promotion programs; intervening early
Service system	Youth	Dealing with family, home and community environments	None reported	None reported
	Service provider	Family socio-economic stressors; Fragmentation across the service system	Working together	Advocating to address the social determinants

## Data Availability

The data presented in this study are available within the article.

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
