# Peer review of "The Availability, Appropriateness, and Integration of Services to Promote Indigenous Australian Youth Wellbeing and Mental Health: Indigenous Youth and Service Provider Perspectives"

_ijerph, 2022, doi:10.3390/ijerph20010375_

Round 1
Reviewer 1 Report
This article is framed in the context of an urgent need for new responses in service provision responses for Indigenous youth mental health support and a need to create best practice regarding community mental health related services by understanding the stakeholders’ perspectives, namely youth and the people who work with the youth. The literature review provides the context for the need, in terms of the survey that shows respondents reporting mental health distress but no intention to seek support; and that health seeking is linked to better outcomes.
The justification for the study is clear, and well argued. The methods are appropriate, the findings important and strong. A particular strength of the article is in bringing the two stakeholders together, namely, the young people who use services and the service providers. However, this does add a complication to the design and the analysis. Relatedly, at times, the description of the design wasn’t clear, and the analysis feels like it might benefit from further iteration. Part of this lack of clarity came from the writing, throughout they were very long sentences which made it hard to follow the points being made by the authors. I suggest in the final edit there’s another careful read-through with a particular focus on simplifying complex sentences.
I outlined these issues below, with a view to suggesting directions for how the authors might maximise the potential of this article.
Literature review:
The literature review focused on identifying the needs and justification of the study. However it didn’t engage with the wider literature on health services for Indigenous youth. Therefore the findings are decontextualised from existing knowledge, limiting how the discussion could develop (see comments about the discussion). Therefore, I suggest some discussion of the literature in terms of what we already know about what young people and service providers think is needed for mental health/ well-being provision for (isolated) indigenous communities.
Method:
I found a description of the design unclear in places. For example, the section (lines 173-6) was unclear to me how the community advisory group was established, how this group differed from the youth yarning circle. Only information about the community advisory group recruitment was given I think? And I was confused as to who participated when and in what. Perhaps separate out section 2.2 into two sections, one with the young people, and one with the service providers and adult community members, so that it is clearer how these two separate groups of people were recruited, what they engaged with for the study, and then have a date it was analysed separately, before the analysis of these two groups were brought into dialogue.
I would like some comment about what kind of grounded theory is being used, there are a range from realist to social constructionist approaches, would be good to know where analysis sits within these possibilities for better understanding of the data analysis.
Analysis
It’s good to avoid one word/topic themes because they tell your reader what is important (e.g. safety) but not how or why. This not only develops the understanding of the content of your theme, but also helps avoid ambiguity, for example it’s hard to know what improvement strategy of “promoting wellness” might be (it could mean so many different things to different people).
I got really confused with the “core issues” sections. I originally thought they were summary sections describing the core issues of the analysis so far, then I understood them being the introduction to the new theme. This confusion was further complicated with relatively long extracts being included, so that I was unclear how these extracts fitted into the actual themes of that section. I therefore recommend taking data out of the “core issues” sections, and perhaps make it clear what are the concept of core issues is, are they the over arcing description of the theme that will follow? Or something else?
I also recommend developing the analysis sections so that they more strongly, and conceptually, engage with the findings of the study. Currently, there is a blurring across some themes, and a flattening out of some of the richness of the data.
The analysis is divided into sections on context, strategies, conditions, strategies, which maps onto the figures. However, the analysis sections never refer back to those figures, and by separating out the analysis into those subheadings, there is repetition (when issues connect to each other across those sections) creating, at times, a ‘bitty’ analysis, because by listing all the problem themes first, and then listing their solutions, the arguments being developed in the paper are scattered, and because of that, less developed than they might be.
My feeling is that the figure works really nicely, but perhaps the data analysis doesn’t have to be in these subheadings but rather bring together issues that relate to each other (for example, that residents experience trauma, and that services for dealing with trauma are needed - so if example, for the Service providers, you might have a theme of “intergenerational and personal trauma” that describes the trauma they see being experienced by these communities and the solutions that these Service providers suggest in relation to that trauma). If the analysis does this while referring to the model/figure, it will be able to discuss these related issues together, while also demonstrating that they are different aspects of the experience (for example, context and strategy).
Some commentary about the figure is also needed, and/or it needs developing. Because although it was a good, visual presentation of findings, summarising a lot of complex information, it was a bit unclear as to what it was communicating. This is because at the core was ‘ feeling reluctant to seek help’, yet strategies around it were, for example, “having safe and available points of contact” - it felt like a contradiction, and instead should logically read that not having safe and available contact would surround a core experience of feeling reluctant to seek help.
I think there is also room to develop the themes themselves for a tighter grounded theory analysis. I think that it would only be a small amount of work to really strengthen the section on the young people’s perspectives. I list these issues below:
Sometimes the data and the analysis don’t map on as clearly as I would like. For example, in the section on connecting with supports and services, an example is given where a young person complains that the police citizen youth club is too limited in terms of time and not being offered to girls. It gets described in the analysis as lacking relevance or a stop start approach, but it seems to me, that this is an example of a continuing but too limited service (i.e. it is relevant, and it is continuing, there just isn’t enough of it).
The theme ‘supports and services (or not)’ has a lot of different concepts in it - there’s talk about having people to go to, not having people to go to, the risks of disclosure, facilities that work and don’t work, terrible experiences when using services. So, I think you need a subheading that better connects with this complexity or have different subheadings that separate out these issues. Maybe distinguish two themes, 1) the things that young people say work, and a bit about the nature of how they work, e.g. that people have somewhere to go that they trust, that they enjoy (eg PCYC), that they feel work for specific issues eg PCYC for feeling stressed, Men Space for feeling down. So perhaps a little bit more analysis on this first. And then a second and different theme in its own right, exploring the issues that limit young people’s ability to access or reach out to services, in terms of how they are limited, inadequate, or even damaging (creating a negative feedback cycle so they won’t use services again, e.g. the hospital services story)
I’m also wondering about the data presented in section 3.1.2, there seems to be data that is relevant for earlier themes, so I’m wondering why all the data doesn’t go into the analysis before 3.1.2, and then 3.1.2 “core issue” exception would summarises and develop the researchers’ analysis of the core issues, rather than introducing more data. (For example, discussion of how all the participants had friends who struggled with mental health issues but couldn’t talk about them openly would go in the theme “hiding mental health concerns”. Doing this might then enable the researchers to explore in more depth the nature of the experiences that the participants are sharing in their talk that it’s coming under the theme of “hiding mental health concerns”. Note that this suggestion was made before I realised that “core issues” is, I think, the introduction to the next theme. The fact that I thought it matched onto the previous theme is a sign that more analytic work needs doing so that each theme is much more clearly distinguished from each other.
RE: Section 3. 1.3, it’s often good to avoid one word themes because they tell your reader what is important (e.g. safety) but not how or why. Wondering if the researchers might want to develop the names of their themes to give a richer sense of the meanings they contain. I’m also interested in how the conclusion for the theme of ‘safety’ was about mental health workers needing strong listening skills, but the theme seemed to be about how being part of a complex social network - against a backdrop of fears of racist mainstream services (doctors/police) - creates a feeling that accessing services isn’t safe (either because it’s not confidential or because it won’t meet your needs/will harm you). Indeed, the conclusion of the theme safety (around strong listening skills etc health workers) mapped better with the following theme of “relationality”. Again, this represents a blurring across themes, and a good theme in grounded theory is one that is distinct from other themes.
I’m also questioning the title of the sub themes in terms of how they map onto the data presented, so for example the theme “awareness of support services” seem to be more about the fact that they were aware of services but that these were perceived as inadequate. This is important because a theme of ‘awareness of support services’ would connect to a solution of creating greater awareness, but if you’re finding that they are aware of the services and they don’t use them because they find them inadequate, then telling them more about the services is not a good solution. This subtheme needs to be developed a bit more, is it that they need awareness, which is implied by that title, or is it that awareness isn’t enough in the context of the services being provided?
Throughout it would be good for the analysis to be developed by focusing more on the data rather than just presenting it. For example, the theme “relationality” could discuss how going to somebody you don’t know is understood as beneficial when contextualised within living in a complex society where everyone might know your business and judge you (i.e. the relationality example of wanting a stranger only happens in the context of the two other themes that you’ve outlined above (namely, not feeling safe and hiding mental health issues as a cultural practice).
I also noticed that there were a couple of examples where the participant’s quotes were talking specifically about gender, (the police club and the netball for ladies quotes), but the analysis ignores this gendered element. In the data that has been presented, it looks like services supporting young women are being asked for, and that seems an important element to recognise?
In the service provider section, the overarching theme of family socio-economic stressors, includes generational trauma and contemporary finances/employment, and concludes with social and economic determinants of health. It would be strengthened if, the literature had discussed social determinants of health, including culture and community. Also the third theme is about “experiences of personal trauma”, so it seems like perhaps a stronger theme be to bring all the trauma discussion together?
The second theme “transitions in education and employment” is it really flat theme in comparison to the content being shared, which seems to be about multiple barriers to employment. If you frame the importance of schooling as feeding into employment, then this theme describes a whole set of barriers to employment, i.e. even though they want it and they identify with it as a desirable transition into adult hood, it’s hard to do it if they haven’t had schooling, don’t have the skills to go out of the community to seek work, and if they do, become vulnerable in terms of housing and losing their family connection/Indigenous ways of being (that provide wellbeing).
The third theme "experiences of personal trauma" has two examples of resilience and ability to support each other through the challenges they face. I appreciate that this maybe to avoid a negative framing of this community but it makes for an unfocused theme. Perhaps, if you move personal experience of trauma talk into a trauma theme, here you can focus on ‘Community solutions from a worker perspective’ – and how they recognise that there is resilience in the community, and mutual support, yet because they don’t have the resources to deal with the kinds of stresses they have to face (like six deaths in a row) they turn to other tools to support themselves like alcohol which can add to the harm. Also that when the community feels overwhelmed, then the community workers feel overwhelmed trying to support them; especially in the context where there already is a low supply of services and supports (and then include the quote from the theme called “supply of services and supporters” that only has one extract in it and so might better work being incorporated into another theme.
The theme “leadership commitment and workforce capacity “seemed to be two separate things, - first, there was an issue about service providers about school principles, health service and other leaders being important, but that didn’t get developed. And then a second issue about workforce capacity, which seem to be the focus of this theme. I suggest either developing the leadership commitment argument e.g. with data, or deleting it and focusing clearly on the topic which connects the quotes presented, namely workforce capacity struggles.
It also might work better not to do some contrasting in the analysis of the service providers, because the arguments get really complicated. Maybe just focus on what the service user’s analysis says, and then in the discussion you can Focus on bringing these two sets of analysis together for a clear comparison.
Discussion.
For clarity, I wonder whether table 3 could be developed so that the shared themes between the youth and the service providers were on the same row, and issues that weren’t shared be given their own separate row. Given that even when there are shared themes the words used to describe these themes are different, only giving the shared themes a shared row would allow a really quick visual communication of what was shared and what was different for the context, core, conditions, strategies.
I think I would also like the discussion section to more clearly be structured around: 1) explicitly answering the research questions, 2) describing the implications for the service provision in this context, 3) describing how these findings feed into wider relevant literature, and therefore identify what may be, and what may not be, transferable to other contexts. Followed by 4) a discussion of limitations, 5) future research, and the final conclusion section.
In relation to point 3, currently, all of this analysis is sitting out of context of the broader research literature, which I think provides support for a lot of these findings, so in the next iteration it would be good to examine how these findings map on (support, develop, or perhaps even contradict) previous research looking at mental health services for indigenous youth – for your readers who might want to apply your findings to their own context, and for your own context because you can see what might be highly context specific and what might be transferable or more strongly evidence-based findings.
Small points:
Typo in the abstract, first two words have got capital letters
Line 181 – ‘they and are a’ should read ‘they are a’
Also super long sentence is a hard to follow, for clarity I recommend most sentences be no more than 25 words, this one for example, the one lines 68-71 is over 80 words long.
Relatedly, sentences with two issues, such as the one on lines 644-5, are hard to read. Suggest going through and clarifying by giving a point per sentence, in this example stating what they service providers think, and then in a different sentence, contrasting it to the young people.
As an aside, the identification as resilient theme is also found in research on African American and Black British women. I wonder if the authors are familiar with the critical literature on the “strong black woman stereotype” as adding an extra burden to women of colour? I raise this this only as a point of interest – in terms of the limits communities have in their responses to racism. It was interesting to see how the young people in your study recognised the damage this apparently affirmative identity can have.
Author Response
8 Dec 2022
Dear Reviewer 1,
A heartfelt thanks for your detailed and insightful comments and suggestions.
Our responses to your comments are provided in red after each paragraph suggestion.
This article is framed in the context of an urgent need for new responses in service provision responses for Indigenous youth mental health support and a need to create best practice regarding community mental health related services by understanding the stakeholders’ perspectives, namely youth and the people who work with the youth. The literature review provides the context for the need, in terms of the survey that shows respondents reporting mental health distress but no intention to seek support; and that health seeking is linked to better outcomes.
The justification for the study is clear, and well argued. The methods are appropriate, the findings important and strong. A particular strength of the article is in bringing the two stakeholders together, namely, the young people who use services and the service providers. However, this does add a complication to the design and the analysis. Relatedly, at times, the description of the design wasn’t clear, and the analysis feels like it might benefit from further iteration. Part of this lack of clarity came from the writing, throughout they were very long sentences which made it hard to follow the points being made by the authors. I suggest in the final edit there’s another careful read-through with a particular focus on simplifying complex sentences.
I outlined these issues below, with a view to suggesting directions for how the authors might maximise the potential of this article.
Literature review: The literature review focused on identifying the needs and justification of the study. However it didn’t engage with the wider literature on health services for Indigenous youth. Therefore the findings are decontextualised from existing knowledge, limiting how the discussion could develop (see comments about the discussion). Therefore, I suggest some discussion of the literature in terms of what we already know about what young people and service providers think is needed for mental health/ well-being provision for (isolated) indigenous communities.
The introduction section has been revised throughout, with the evidence from additional studies included to contextualise the study.
Method: I found a description of the design unclear in places. For example, the section (lines 173-6) was unclear to me how the community advisory group was established, how this group differed from the youth yarning circle. Only information about the community advisory group recruitment was given I think? And I was confused as to who participated when and in what. Perhaps separate out section 2.2 into two sections, one with the young people, and one with the service providers and adult community members, so that it is clearer how these two separate groups of people were recruited, what they engaged with for the study, and then have a date it was analysed separately, before the analysis of these two groups were brought into dialogue.
Participants in community youth advisory groups and yarning circles were both recruited primarily through invitations from the Gurriny Youth Wellbeing Officers – given the small size of the community, these youth wellbeing officers were familiar with youth interests and capabilities (See Lines 204-206). The main difference between the yarning circle and CYAG participants was their age – yarning group was with younger youth (11-16 years). The CYAG meetings were with older youth (16-25 years). This is clarified in the paper Line 196 and 203. The tables of participants have been edited for further clarity, including the dates of meetings/data collection. Line 234 and Line 250.
I would like some comment about what kind of grounded theory is being used, there are a range from realist to social constructionist approaches, would be good to know where analysis sits within these possibilities for better understanding of the data analysis.
The type of grounded theory (Social Constructivist) is now identified Line 134
Analysis
It’s good to avoid one word/topic themes because they tell your reader what is important (e.g. safety) but not how or why. This not only develops the understanding of the content of your theme, but also helps avoid ambiguity, for example it’s hard to know what improvement strategy of “promoting wellness” might be (it could mean so many different things to different people).
The results section has been revised and restructured throughout. As part of this process, the terms used for subheadings have been reviewed, with one word subheading amended and all terms assessed to check clarity.
I got really confused with the “core issues” sections. I originally thought they were summary sections describing the core issues of the analysis so far, then I understood them being the introduction to the new theme. This confusion was further complicated with relatively long extracts being included, so that I was unclear how these extracts fitted into the actual themes of that section. I therefore recommend taking data out of the “core issues” sections, and perhaps make it clear what are the concept of core issues is, are they the over arcing description of the theme that will follow? Or something else?
The core issues section has been moved to the start of the results section, a new joint “core” established and the extracts reduced as suggested. We trust that this makes the purpose of the core process clearer, Line 295-340.
I also recommend developing the analysis sections so that they more strongly, and conceptually, engage with the findings of the study. Currently, there is a blurring across some themes, and a flattening out of some of the richness of the data.
As above, the results section has been restructured throughout to consolidate and clarify the key arguments. The new structure provides the results according to the level of focus of the context, conditions and strategies – i.e. at the level of youth, families, services or the service system. This new structure is signposted by the new Table 3 Line 293. The intent is to provide a coherent argument across each (from the perspective of youth and service providers) and identify Yarrabah improvement strategies at each level of focus going forward.
The analysis is divided into sections on context, strategies, conditions, strategies, which maps onto the figures. However, the analysis sections never refer back to those figures, and by separating out the analysis into those subheadings, there is repetition (when issues connect to each other across those sections) creating, at times, a ‘bitty’ analysis, because by listing all the problem themes first, and then listing their solutions, the arguments being developed in the paper are scattered, and because of that, less developed than they might be.
As above – we have attempted to pull together the youth and service providers’ perspectives about the issues at the various levels (whilst holding true to their distinct perspectives based on their positioning). The two diagrams have been summarised in Table 3 (Line 293) and combined into one overarching diagram in discussion (Line 861). The overarching headings are now youth, families, services, and service system.
My feeling is that the figure works really nicely, but perhaps the data analysis doesn’t have to be in these subheadings but rather bring together issues that relate to each other (for example, that residents experience trauma, and that services for dealing with trauma are needed - so if example, for the Service providers, you might have a theme of “intergenerational and personal trauma” that describes the trauma they see being experienced by these communities and the solutions that these Service providers suggest in relation to that trauma). If the analysis does this while referring to the model/figure, it will be able to discuss these related issues together, while also demonstrating that they are different aspects of the experience (for example, context and strategy).
We agree and have made this change – as suggested.
Some commentary about the figure is also needed, and/or it needs developing. Because although it was a good, visual presentation of findings, summarising a lot of complex information, it was a bit unclear as to what it was communicating. This is because at the core was ‘ feeling reluctant to seek help’, yet strategies around it were, for example, “having safe and available points of contact” - it felt like a contradiction, and instead should logically read that not having safe and available contact would surround a core experience of feeling reluctant to seek help.
The diagram has been amended as above into Table 3 and Figure 2. The enablers and strategies outline what is needed to deal with the youth core reluctance to seek help. Reluctance to seek help is a logical outcome of the incongruence of supports and services to meet youth needs.
I think there is also room to develop the themes themselves for a tighter grounded theory analysis. I think that it would only be a small amount of work to really strengthen the section on the young people’s perspectives. I list these issues below:
Sometimes the data and the analysis don’t map on as clearly as I would like. For example, in the section on connecting with supports and services, an example is given where a young person complains that the police citizen youth club is too limited in terms of time and not being offered to girls. It gets described in the analysis as lacking relevance or a stop start approach, but it seems to me, that this is an example of a continuing but too limited service (i.e. it is relevant, and it is continuing, there just isn’t enough of it).
The text has been changed to focus on the limited capacity of services to meet the needs of Yarrabah youth. Line 630.
The theme ‘supports and services (or not)’ has a lot of different concepts in it - there’s talk about having people to go to, not having people to go to, the risks of disclosure, facilities that work and don’t work, terrible experiences when using services. So, I think you need a subheading that better connects with this complexity or have different subheadings that separate out these issues. Maybe distinguish two themes, 1) the things that young people say work, and a bit about the nature of how they work, e.g. that people have somewhere to go that they trust, that they enjoy (eg PCYC), that they feel work for specific issues eg PCYC for feeling stressed, Men Space for feeling down. So perhaps a little bit more analysis on this first. And then a second and different theme in its own right, exploring the issues that limit young people’s ability to access or reach out to services, in terms of how they are limited, inadequate, or even damaging (creating a negative feedback cycle so they won’t use services again, e.g. the hospital services story)
This section has been amended to deal just with which services have been utilized by youth. Lines 630-44.
I’m also wondering about the data presented in section 3.1.2, there seems to be data that is relevant for earlier themes, so I’m wondering why all the data doesn’t go into the analysis before 3.1.2, and then 3.1.2 “core issue” exception would summarises and develop the researchers’ analysis of the core issues, rather than introducing more data. (For example, discussion of how all the participants had friends who struggled with mental health issues but couldn’t talk about them openly would go in the theme “hiding mental health concerns”. Doing this might then enable the researchers to explore in more depth the nature of the experiences that the participants are sharing in their talk that it’s coming under the theme of “hiding mental health concerns”. Note that this suggestion was made before I realised that “core issues” is, I think, the introduction to the next theme. The fact that I thought it matched onto the previous theme is a sign that more analytic work needs doing so that each theme is much more clearly distinguished from each other.
As above – we fore fronted the core issues section, reduced its length and honed the included concepts Lines 295-344.
RE: Section 3. 1.3, it’s often good to avoid one word themes because they tell your reader what is important (e.g. safety) but not how or why. Wondering if the researchers might want to develop the names of their themes to give a richer sense of the meanings they contain. I’m also interested in how the conclusion for the theme of ‘safety’ was about mental health workers needing strong listening skills, but the theme seemed to be about how being part of a complex social network - against a backdrop of fears of racist mainstream services (doctors/police) - creates a feeling that accessing services isn’t safe (either because it’s not confidential or because it won’t meet your needs/will harm you). Indeed, the conclusion of the theme safety (around strong listening skills etc health workers) mapped better with the following theme of “relationality”. Again, this represents a blurring across themes, and a good theme in grounded theory is one that is distinct from other themes.
We combined the issues of safety with trust, relationality and consistency of service providers. These were the values that youth identified as being important in their encounters with mental health services, and it was hard to tease them apart. For example, relationality is a component of cultural safety and vice versa. Lines 425-460.
I’m also questioning the title of the sub themes in terms of how they map onto the data presented, so for example the theme “awareness of support services” seem to be more about the fact that they were aware of services but that these were perceived as inadequate. This is important because a theme of ‘awareness of support services’ would connect to a solution of creating greater awareness, but if you’re finding that they are aware of the services and they don’t use them because they find them inadequate, then telling them more about the services is not a good solution. This subtheme needs to be developed a bit more, is it that they need awareness, which is implied by that title, or is it that awareness isn’t enough in the context of the services being provided?
Agree – the term has been changed to adequacy and appropriateness of support services Line 630.
Throughout it would be good for the analysis to be developed by focusing more on the data rather than just presenting it. For example, the theme “relationality” could discuss how going to somebody you don’t know is understood as beneficial when contextualised within living in a complex society where everyone might know your business and judge you (i.e. the relationality example of wanting a stranger only happens in the context of the two other themes that you’ve outlined above (namely, not feeling safe and hiding mental health issues as a cultural practice).
We attempted to find a balance between staying true to participants voices and interpretations, and our analysis of the data. Given the multi-level analysis of an already complex issue, and our concern about an already lengthy paper, we made the decision to preference participants’ voices.
I also noticed that there were a couple of examples where the participant’s quotes were talking specifically about gender, (the police club and the netball for ladies quotes), but the analysis ignores this gendered element. In the data that has been presented, it looks like services supporting young women are being asked for, and that seems an important element to recognise?
There were data focussed on gender-related issues; some about the need for additional activities for girls/young women and some about boys/young men. There were also some data that specified age-specific suggestions. We have not highlighted this diversity in the analysis because the intent was to develop an overview of youth and service providers’ perspectives of the Yarrabah situation to inform improvement strategies – once we have honed these, it is possible that gender-based activities are requested (these are not yet determined).
In the service provider section, the overarching theme of family socio-economic stressors, includes generational trauma and contemporary finances/employment, and concludes with social and economic determinants of health. It would be strengthened if, the literature had discussed social determinants of health, including culture and community. Also the third theme is about “experiences of personal trauma”, so it seems like perhaps a stronger theme be to bring all the trauma discussion together?
Amended as suggested, Line 384.
The second theme “transitions in education and employment” is it really flat theme in comparison to the content being shared, which seems to be about multiple barriers to employment. If you frame the importance of schooling as feeding into employment, then this theme describes a whole set of barriers to employment, i.e. even though they want it and they identify with it as a desirable transition into adult hood, it’s hard to do it if they haven’t had schooling, don’t have the skills to go out of the community to seek work, and if they do, become vulnerable in terms of housing and losing their family connection/Indigenous ways of being (that provide wellbeing).
Agreed – this content has been combined with the category about social determinants of health under the heading “family socio economic stressors Line 772.
The third theme "experiences of personal trauma" has two examples of resilience and ability to support each other through the challenges they face. I appreciate that this maybe to avoid a negative framing of this community but it makes for an unfocused theme. Perhaps, if you move personal experience of trauma talk into a trauma theme, here you can focus on ‘Community solutions from a worker perspective’ – and how they recognise that there is resilience in the community, and mutual support, yet because they don’t have the resources to deal with the kinds of stresses they have to face (like six deaths in a row) they turn to other tools to support themselves like alcohol which can add to the harm. Also that when the community feels overwhelmed, then the community workers feel overwhelmed trying to support them; especially in the context where there already is a low supply of services and supports (and then include the quote from the theme called “supply of services and supporters” that only has one extract in it and so might better work being incorporated into another theme.
Amended as suggested, Line 384.
The theme “leadership commitment and workforce capacity “seemed to be two separate things, - first, there was an issue about service providers about school principles, health service and other leaders being important, but that didn’t get developed. And then a second issue about workforce capacity, which seem to be the focus of this theme. I suggest either developing the leadership commitment argument e.g. with data, or deleting it and focusing clearly on the topic which connects the quotes presented, namely workforce capacity struggles.
Amended as suggested Line 645.
It also might work better not to do some contrasting in the analysis of the service providers, because the arguments get really complicated. Maybe just focus on what the service user’s analysis says, and then in the discussion you can Focus on bringing these two sets of analysis together for a clear comparison.
Agreed – and amended throughout as suggested.
Discussion.
For clarity, I wonder whether table 3 could be developed so that the shared themes between the youth and the service providers were on the same row, and issues that weren’t shared be given their own separate row. Given that even when there are shared themes the words used to describe these themes are different, only giving the shared themes a shared row would allow a really quick visual communication of what was shared and what was different for the context, core, conditions, strategies.
Amended as suggested and moved to the results section, Line 293.
I think I would also like the discussion section to more clearly be structured around: 1) explicitly answering the research questions, 2) describing the implications for the service provision in this context, 3) describing how these findings feed into wider relevant literature, and therefore identify what may be, and what may not be, transferable to other contexts. Followed by 4) a discussion of limitations, 5) future research, and the final conclusion section.
The discussion has been revised as suggested.
In relation to point 3, currently, all of this analysis is sitting out of context of the broader research literature, which I think provides support for a lot of these findings, so in the next iteration it would be good to examine how these findings map on (support, develop, or perhaps even contradict) previous research looking at mental health services for indigenous youth – for your readers who might want to apply your findings to their own context, and for your own context because you can see what might be highly context specific and what might be transferable or more strongly evidence-based findings.
As above
Small points: Typo in the abstract, first two words have got capital letters
Line 181 – ‘they and are a’ should read ‘they are a’
Corrected
Also super long sentence is a hard to follow, for clarity I recommend most sentences be no more than 25 words, this one for example, the one lines 68-71 is over 80 words long.
The paper has been edited throughout with length of sentences checked and repaired.
Relatedly, sentences with two issues, such as the one on lines 644-5, are hard to read. Suggest going through and clarifying by giving a point per sentence, in this example stating what they service providers think, and then in a different sentence, contrasting it to the young people.
Amended.
As an aside, the identification as resilient theme is also found in research on African American and Black British women. I wonder if the authors are familiar with the critical literature on the “strong black woman stereotype” as adding an extra burden to women of colour? I raise this this only as a point of interest – in terms of the limits communities have in their responses to racism. It was interesting to see how the young people in your study recognised the damage this apparently affirmative identity can have.
Thanks – very much appreciate your very insightful, thorough and considered comments and suggestions.
Yours, for the author team,
Janya McCalman
Reviewer 2 Report
This paper present very important and useful data. However the current manuscript would require extensive revisions to be ready for publication.
Introduction- the introduction provides some important background information. However, some of the statistics that are reported are very detailed. A more broad overview of Australian Indigenous based mental health disparities would be useful. The overall rationale for the study remained unclear to me. This was in part related to the use of both hypothesis and research aims but also a lack of explicitly identifying the gap in research. It also remained unclear to me how the current model operated in terms of inter-organisational input. This seems relevant given the study sampled cross-sector participants. The overall structure of the introduction needs some modification to improve the flow and readability. There are a number of long awkward sentences and unnecessary use of semi-colons. The paragraphs are long and change direction and topic.
Methods. Some information is missing. How were the participants selected, approached and recruited. More detail is required regarding the service provider participants. The introduction signposts this is a cross sector study; however, only two non-health organisation participants were involved. This needs clarity. Line 208 states that the yarning circles were augmented with individual consultations- what does this mean? Were individual recorded interviews conducted? Analysis well described.
Results- the overall theme development seems reasonable but the results section relies heavily on raw data to present the results. The results are also very detailed and long and the more important findings are lost in excessive detail. This could be reduced by more definition and interpretation by the authors with less reliance on raw data. Another option would be to present the results across two papers or to condense them down considerably.
Discussion
I would be cautious about framing the results around the idea of Indigenous youth being reluctant to seek help. This places the onus on them rather than viewing the historical and institutional context in which enablers and barriers to treatment occur (which you have also discussed). Equally this relates to the idea of intergenerational trauma, where by the use of this term indicates the issue is centered within the families rather than focusing on the societal, systemic and institutional factors that have maintained or at least not interrupted the consequences of colonisation.
The discussion would benefit from more consideration of the implications of the findings. So what does this mean in terms of current models of Indigenous mental health delivery, what would need to change.
Author Response
8 Dec 2022
Dear Reviewer 2,
A heartfelt thanks for your detailed and insightful comments and suggestions.
Our responses to your comments are provided in red after each paragraph suggestion.
This paper present very important and useful data. However the current manuscript would require extensive revisions to be ready for publication.
Introduction- the introduction provides some important background information. However, some of the statistics that are reported are very detailed.
The introduction has been revised; we have summarized the reported statistics e.g. Line 68.
A more broad overview of Australian Indigenous based mental health disparities would be useful.
The focus of the paper is on community-driven improvements to the wellbeing/mental health service system rather than mental health disparities per se. Indigenous researchers have requested that the discourse about Indigenous people NOT focus on deficits. Instead, we included data about the high levels of Indigenous youth concern about mental health and the high levels of stress experienced. This was included to provide some context to the significance of the study. We also added information about disparities in hospitalization for mental illness Line 64-65.
The overall rationale for the study remained unclear to me. This was in part related to the use of both hypothesis and research aims but also a lack of explicitly identifying the gap in research.
We have clarified the rationale for the paper in the abstract (lines 14-20) and introduction (Lines 49-52) – that whilst policy documents recommend community-based solutions to the fragmentation of youth mental health services, there is little evidence of how Indigenous communities are coming together to improve service system integration. This paper provides a case example of initial efforts in one discrete Indigenous community to identify the issues, enabling conditions and improvement strategies – from the perspectives of youth and service providers. The hypotheses have been removed.
It also remained unclear to me how the current model operated in terms of inter-organisational input. This seems relevant given the study sampled cross-sector participants.
A community perspective was taken in the study – the model is relevant for youth-health related organisations in Yarrabah.
The overall structure of the introduction needs some modification to improve the flow and readability. There are a number of long awkward sentences and unnecessary use of semi-colons. The paragraphs are long and change direction and topic.
The entire paper has been revised and edited to improve flow and readability. We have attended to the lengthy sentences and unwieldy paragraphs.
Methods. Some information is missing. How were the participants selected, approached and recruited.
Amended to clarify that the recruitment strategies via local Gurriny youth wellbeing officers who were familiar with the youth participants and service providers (Yarrabah is a discrete community of ~2500 people so relationships within the community are close) Line 204-206.
More detail is required regarding the service provider participants. The introduction signposts this is a cross sector study; however, only two non-health organisation participants were involved. This needs clarity.
As the only primary healthcare service in Yarrabah and having a holistic focus, Gurriny is responsible for services that would probably not be the case in mainstream towns e.g. a drop in youth hub. So while only two non-health organisation participants were involved, the key players representing youth wellbeing activities and services for Gurriny, the school and alcohol treatment service were there.
Line 208 states that the yarning circles were augmented with individual consultations- what does this mean? Were individual recorded interviews conducted?
Amended to use the term interviews, line 237. These were recorded.
Analysis well described.
Thanks
Results- the overall theme development seems reasonable but the results section relies heavily on raw data to present the results. The results are also very detailed and long and the more important findings are lost in excessive detail. This could be reduced by more definition and interpretation by the authors with less reliance on raw data. Another option would be to present the results across two papers or to condense them down considerably.
We attempted to find a balance between staying true to participants voices and interpretations, and our analysis of the data. Given the multi-level analysis of an already complex issue, and our concern about an already lengthy paper, we made the decision to preference participants’ voices.
We have restructured the results section throughout to provide more clarity about the more important findings.
Discussion
I would be cautious about framing the results around the idea of Indigenous youth being reluctant to seek help. This places the onus on them rather than viewing the historical and institutional context in which enablers and barriers to treatment occur (which you have also discussed). Equally this relates to the idea of intergenerational trauma, where by the use of this term indicates the issue is centered within the families rather than focusing on the societal, systemic and institutional factors that have maintained or at least not interrupted the consequences of colonisation.
Agreed – youth were reluctant to seek help because the services and supports were incongruent to their needs. We have revised the entire results section and hope that we have clarified that are not putting the onus on them – but rather representing their view that they are reluctant to seek help because the system is not working for them (an entirely rational response on their part – albeit one with unfortunate consequences).
The discussion would benefit from more consideration of the implications of the findings. So what does this mean in terms of current models of Indigenous mental health delivery, what would need to change.
The discussion has also been revised. The next steps identified are to implement community-defined strategies for improvement.
Thank you for your comments and suggestions – we very much appreciate your time and thoughtfulness.
Yours, for the author team,
Janya McCalman